# Generation and Characterization of a New Preclinical Mouse Model of EGFR-Driven Lung Cancer with MET-Induced Osimertinib Resistance

**DOI:** 10.3390/cancers13143441

**Published:** 2021-07-09

**Authors:** Maicol Mancini, Quentin-Dominique Thomas, Sylvia Bourdel, Laura Papon, Emilie Bousquet, Prisca Jalta, Silvia La Monica, Camille Travert, Roberta Alfieri, Xavier Quantin, Marta Cañamero, Antonio Maraver

**Affiliations:** 1Oncogenic Pathways in Lung Cancer, Institut de Recherche en Cancérologie de Montpellier (IRCM)-Université de Montpellier (UM)-Institut Régional du Cancer de Montpellier (ICM), CEDEX 5, F-34298 Montpellier, France; Quentin.Thomas@icm.unicancer.fr (Q.-D.T.); sylvia.bourdel@inserm.fr (S.B.); Laura.Papon@icm.unicancer.fr (L.P.); emilie.bousquet46@gmail.com (E.B.); prisca.jalta@etu.umontpellier.fr (P.J.); Camille.Travert@icm.unicancer.fr (C.T.); xavier.quantin@icm.unicancer.fr (X.Q.); 2Laboratorio di Oncologia Sperimentale, Dipartimento di Medicina e Chirurgia, Università di Parma, 43126 Parma, Italy; silvia.lamonica@unipr.it (S.L.M.); roberta.alfieri@unipr.it (R.A.); 3Roche Innovation Center, Roche Pharmaceutical Research and Early Development, 82377 Penzberg, Germany; Marta.canamero@roche.com

**Keywords:** lung cancer, EGFR, MET, TKI, preclinical mouse models

## Abstract

**Simple Summary:**

The use of targeted therapy has changed the clinical management of lung cancer patients, increasing both their life quality and expectancy. Conversely, the appearance of resistance occurs in almost all patients receiving this therapy. In this regard, new strategies combining different therapies could delay or even eliminate the appearance of resistance. However, in order to develop new therapeutic treatments, we need preclinical mouse models that recapitulate human disease. In the present study, we developed a new state-of-the-art mouse model that summarizes all features occurring in EGFR-mutated patients that relapse after osimertinib after acquisition of MET amplification.

**Abstract:**

Despite the introduction of epidermal growth factor receptor (EGFR) tyrosine kinase inhibitors (TKIs) to treat advanced lung cancer harboring EGFR-activating mutations, the prognosis remains unfavorable because of intrinsic and/or acquired resistance. We generated a new state-of-the-art mouse strain harboring the human EGFR^T790M/L858R^ oncogene and MET overexpression (EGFR/MET strain) that mimics the MET amplification occurring in one out of five patients with EGFR-mutated lung cancer that relapsed after treatment with osimertinib, a third-generation anti-EGFR TKI. We found that survival was reduced in EGFR/MET mice compared with mice harboring only EGFR^T790M/L858R^ (EGFR strain). Moreover, EGFR/MET-driven lung tumors were resistant to osimertinib, recapitulating the phenotype observed in patients. Conversely, as also observed in patients, the crizotinib (anti-MET TKI) and osimertinib combination improved survival and reduced tumor burden in EGFR/MET mice, further validating the model’s value for preclinical studies. We also found that in EGFR/MET mice, MET overexpression negatively regulated EGFR activity through MIG6 induction, a compensatory mechanism that allows the coexistence of the two onco-genic events. Our data suggest that single EGFR or MET inhibition might not be a good therapeutic option for EGFR-mutated lung cancer with MET amplification, and that inhibition of both pathways should be the best clinical choice in these patients.

## 1. Introduction

EGFR-activating mutations, such as exon 19 deletion or the L858R mutation in exon 21, occur in 10–15% of patients with non-small cell lung cancer. For these patients, targeted therapy is proposed as first-line treatment because it provides a clear advantage compared to standard chemotherapy. First-generation (e.g., gefitinib and erlotinib) and second-generation (e.g., afatinib) tyrosine kinase inhibitors (TKIs) can be used as first-line treatment in these patients, but resistance invariably appears in all patients. In 50% to 65% of them, this occurs through acquisition of the so-called gatekeeper EGFR mutation T790M [1]. Interestingly, third-generation TKIs (i.e., osimertinib) led to important tumor regression in EGFR^T790M^ lung cancer following relapse after first-generation TKIs [2]. Moreover, in the FLAURA phase 3 clinical trial, progression-free survival and overall survival were longer with osimertinib than with first-generation TKIs and, therefore, in many countries, osimertinib has become the first-line standard of care for patients with EGFR-mutated non-small cell lung cancer (NSCLC) [3]. Nevertheless, relapse also occurs in osimertinib-treated patients. Several studies showed that in patients with EGFR-mutated lung cancer, osimertinib resistance can be promoted by MET amplification in around 20% of cases [4]. A very recent phase 1b clinical trial showed a therapeutic benefit of the combination of osimertinib and the MET inhibitor savolitinib in this patient subtype [5]. Although these results are very encouraging, the therapeutic effectiveness of osimertinib needs to be increased because the effect of the MET inhibitor–TKI combination in this particular trial remained modest. Importantly, the implementation of new treatments might be delayed due to the lack of good preclinical tools.

Here, we describe the generation and characterization of a new mouse model obtained by combining doxycycline-inducible expression in the lung of the human EGFR^T790M/L858R^ oncogene, which promotes the development of EGFR-driven lung cancer when expressed alone [6], and wild type human MET, using a transgenic mouse model that promotes liver carcinoma when a liver-specific promoter is used [7]. Our new model fully recapitulates the treatment response observed in patients with EGFR-mutated lung cancer who relapse after osimertinib. We also found that MET overexpression reduced the levels of phosphorylated EGFR (pEGFR, i.e., the activated form) and total EGFR, and this effect was prevented by MET inhibitors. Moreover, the expression of MIG6, a negative regulator of EGFR activity [8], was positively regulated by MET and MIG6 loss of function restored pEGFR and EGFR levels upon MET overexpression, providing a mechanistic hint to this interesting phenomenon.

The present study offers a rationale to propose EGFR and MET inhibitors always in combination to patients with lung cancer harboring EGFR mutations and MET amplification. Our new compound mouse strain will pave the way to in vivo preclinical studies to test new therapeutic options in this specific genetic context with important clinical implications.

## 2. Results

### 2.1. MET Overexpression Decreases Survival of Mice Harboring EGFR-Driven Lung Adenocarcinoma

To develop a mouse model that could recapitulate osimertinib resistance caused by MET amplification in patients with EGFR-mutated lung cancer, we took advantage of a mouse strain in which liver carcinoma develops upon induction of human WT MET expression in the liver [7]. This mouse model relies on the same doxycycline-inducible genetic system used in the EGFR^T790M/L858R^ transgenic mouse strain that we previously crossed with the lung-specific CCSP-rtTA strain (EGFR^T790M/L858R^/rtTA, EGFR mice hereafter) and in which EGFR-driven lung adenocarcinoma appears upon doxycycline exposure [6]. We crossed the MET and EGFR strains to generate the EGFR^T790M/L858R^/MET/rtTA compound mouse (EGFR/MET mice hereafter) (Figure 1A). Upon doxycycline exposure, survival was reduced in EGFR/MET mice compared with EGFR mice (median survival: 18.2 weeks for EGFR and 6.5 weeks for EGFR/MET mice) (Figure 1B). Death was related to breathing problems and we observed tumors only in lungs. Analysis of a subset of moribund mice showed a similar tumor burden in lungs from both strains (Appendix A). However, and in accordance with the survival curves, EGFR/MET mice reached this stage earlier than EGFR mice (Appendix A). Histological analysis of lung samples from these mice showed the same lung adenocarcinoma features in both genotypes (Appendix A): solid pattern with high mitotic activity, nuclear atypia, and presence of cytoplasmic vacuoles in some cases. Moreover, in EGFR/MET-driven tumor samples, there were also extensive areas of alveolar hyperplasia and pneumonia, with alveoli filled with macrophages and plasma cells surrounding the adenocarcinoma mass or within edematous areas in the tumor (Appendix A). The reasons for such differences in infiltration will require further investigations, but they probably contributed to the reduced survival of EGFR/MET mice. Finally, the percentage of adenocarcinomas relative to the adenomas at death was lower in EGFR/MET than EGFR mice (Figure 1C), a feature probably related to their shorter survival with less time to allow tumors to become full-blown carcinomas.

We also analyzed the expression of different proteins implicated in EGFR-driven lung adenocarcinoma. As expected, EGFR/MET-driven tumors showed strong MET expression and activation (phosphorylated MET, pMET) compared with EGFR-driven tumors (Figure 1D). Conversely, phosphorylated ERK (pERK) levels were lower, while phosphorylated AKT (pAKT) levels were similar between genotypes. Intriguingly, EGFR and pEGFR levels were strongly reduced in EGFR/MET-driven tumors compared with EGFR-driven tumors (Figure 1D), suggesting that MET overexpression negatively regulates EGFR.

### 2.2. MET Overexpression Decreases EGFR Activity

To determine whether the MET-induced EGFR inhibition observed in lung tumors from EGFR/MET mice was clinically relevant, we performed a correlation analysis of pEGFR and pMET expression in a TCGA cohort of patients with lung adenocarcinoma [9]. We found a negative correlation between pEGFR and pMET expression, suggesting that active MET could also inhibit EGFR activity in patients (Figure 2A) as it could also suggest that the two populations, i.e., high pEGFR and high pMET, are mutually exclusive.

To investigate the underlying mechanism, we used the human lung adenocarcinoma H1975 cell line that, as in our mouse models, also harbor the EGFR^T790M/L858R^ mutations. We transduced this cell line with lentivirus particles harboring a newly generated doxycycline-inducible MET expression plasmid, obtaining an H1975 cell line in which pMET and total MET were strongly expressed upon exposure to doxycycline (Figure 2B). Of note, pEGFR and total EGFR levels were decreased upon induction of MET expression (Figure 2B), thus recapitulating our in vivo findings (Figure 1D). We also confirmed this observation in another lung adenocarcinoma cell line with inducible MET expression that we generated starting from the KRAS-mutated A549 cell line (Appendix A). Altogether, these findings demonstrated that MET expression decreases EGFR activity.

In order to understand this phenotype, we looked for negative regulators of EGFR and, among the different ones, we focused on MIG6 because (i) it is upregulated by EGFR and in turn modulates EGFR function [8] and (ii) because MIG6 has been reported to also be upregulated by the MET ligand HGF [10]; hence we hypothesized that MIG6 could be affected by MET overexpression (amplification or ectopic expression). Importantly, MIG6 regulates EGFR at the phosphorylation level [10], and the negative correlation we observed in patients was between pEGFR and pMET (Figure 2A). Therefore, we first analyzed, in the same patient cohort, the correlation between EGFR and the gene encoding for MIG6 (*ERRFI1*) expression, and found a positive correlation (Figure 2C, left panel). *ERRFI1* was also positively correlated with *MET* expression, with a higher Pearson correlation coefficient than for the *ERRFI1–EGFR* correlation (Figure 2C, right panel). Then, we analyzed for MIG6 gene mRNA expression in inducible MET-expressing H1975 and A549 cells and found increased mRNA levels upon incubation with doxycycline (Appendix A). In accordance with the increased mRNA levels, MIG6 protein levels were also higher in both cell lines upon MET expression (Appendix A). In order to firmly link MIG6 to the MET-induced EGFR and pEGFR decreased expression, we performed MIG6 loss of function experiments in H1975 and A549. Interestingly, when MIG6 levels were reduced in A549 cells, the levels of EGFR and pEGFR upon MET overexpression were restored when compared to the steady state control, and the same phenotype also occurred in H1975, albeit to a lesser extent (Figure 2D).

Altogether, these data indicated that MET overexpression in human cells increased the expression of MIG6, a negative regulator of EGFR, which in turn diminished pEGFR and EGFR levels.

### 2.3. EGFR/MET-Driven Tumors Are Resistant to Osimertinib

An important feature in patients with EGFR-mutated lung cancer and MET amplification is that they are resistant to osimertinib [4]. To validate our new preclinical model, we treated EGFR and EGFR/MET mice harboring tumors with this TKI. As survival was shorter in EGFR/MET than EGFR mice (Figure 1B), we treated mice with doxycycline for 9 (EGFR) or 5 weeks (EGFR/MET). The EGFR/MET group started doxycycline 4 weeks later than the EGFR group to then initiate the TKI treatment at the same time (see Appendix A for the experiment design). After induction, mice were randomized into two groups to be treated with vehicle or osimertinib for 4 weeks. As before, all vehicle-treated EGFR/MET mice reached the end point (breathing distress and suffocation) before the treatment ended, while only one vehicle-treated EGFR mouse reached this point during this period. All other EGFR mice (vehicle or osimertinib) were euthanized after the 4 weeks of treatment. Analysis of tumor burden showed that in EGFR mice, osimertinib efficiently controlled EGFR tumor growth (Figure 3A,B), with a three-fold decrease in the percentage of tumor cells (52% in vehicle-treated and 17% in osimertinib-treated mice) (Figure 3B). Conversely, in EGFR/MET mice, osimertinib’s effect was mild and non-significant (60% of tumor cells in vehicle-treated and 48% in osimertinib-treated mice) (Figure 3B). This demonstrated that EGFR/MET tumors are resistant to osimertinib, mimicking what is observed in EGFR-mutated lung cancer patients with MET amplification.

Protein expression analysis in these mice highlighted several features. First, and in accordance with our previous data, pEGFR and total EGFR levels were higher in EGFR-driven than EGFR/MET-driven tumors. Then, osimertinib treatment decreased pEGFR level in all tumors, regardless of the genotype (Figure 3C). Osimertinib treatment also decreased pERK and pAKT levels in EGFR-driven but not in EGFR/MET-driven tumors, where pAKT levels even increased (Figure 3C), further confirming that EGFR/MET tumors are resistant to osimertinib. Moreover, osimertinib strongly reduced MIG6 protein and mRNA levels in EGFR-driven but not in EGFR/MET-driven tumors (Figure 3D and Appendix A), probably because MET maintained MIG6 expression. Of note, MIG6 expression was higher in EGFR/MET-driven than EGFR-driven tumors, indicating that, as it happens in human cells, MET increased MIG6 expression.

We conclude that our new mouse strain mimics the osimertinib resistance observed in patients with EGFR-mutated and MET-amplified lung cancer.

### 2.4. EGFR/MET-Driven Tumors Are Sensitive to the Osimertinib and Crizotinib Combination

We sought to inhibit both pathways at the same time and hence we treated EGFR/MET mice with the combination of osimertinib and crizotinib, a clinically relevant MET TKI [11,12]. We exposed EGFR/MET mice to doxycycline for 5 weeks as before (Appendix A), and then randomized them into four groups: vehicle, single treatment (osimertinib or crizotinib) and their combination. We treated mice until all mice reached the end point, and then collected samples for analysis. After 6 weeks of treatment, only animals in the combination group were still alive and we euthanized them to analyze all mice at the same time. One mouse in the combination treatment group died due to a wound caused by another mouse that could not be healed and was censored for the survival analysis.

The osimertinib/crizotinib combination increased survival compared with all other groups (Figure 4A). Similarly, tumor burden was significantly lower in the combination group (23%) compared with vehicle- (55%) and crizotinib-treated mice (53%) (*p* < 0.01). We observed a similar trend compared with osimertinib-treated mice (43%) (*p* = 0.06) (Figure 4B and Appendix A).

Protein expression analysis revealed that crizotinib treatment increased pEGFR expression, confirming the negative effect of MET expression on EGFR (Figure 4C). In this treatment group, pMET was strongly reduced, as expected, and also pAKT and pERK. As before, osimertinib induced a strong decrease in pEGFR, and increased pAKT level. Conversely, pERK was decreased compared with vehicle, possibly due to the longer treatment period (6 weeks vs. 4 weeks in the experiment of Figure 3C). The combination very strongly reduced pMET, pEGFR, pAKT and pERK levels, further confirming that this treatment is the most effective against EGFR/MET-driven tumors. Finally, osimertinib and crizotinib alone did not decrease MIG6 protein or mRNA levels, suggesting that EGFR and MET might compensate each other when only one is inhibited. Indeed, MIG6 levels (protein and mRNA) were decreased only when both pathways were inhibited with the drug combination (Figure 4D and Appendix A).

Finally, we report the case of a 69-year-old woman treated by us in our local hospital (Montpellier Cancer Institute, ICM, Montpellier, France) (Appendix A). She was diagnosed with NSCLC harboring an exon 19 deletion in EGFR. She was treated with afatinib (second-generation EGFR TKI) for 23 months. Pleural progression led to a second line of chemotherapy (platinum-based therapy combined with pemetrexed) for four cycles, resulting in renal failure and hence treatment stopped. Osimertinib was proposed as third-line treatment and initiated 33 months after first diagnosis, concomitant with observation of brain metastasis. Pulmonary tumor biopsy was also performed at this time and we found a MET amplification (MET/CEP7 ratio > 4.08). Osimertinib treatment promoted a complete cerebral response but, conversely, there was progression in abdominal lymph nodes and adrenal glands less than 3 months after treatment initiation. Since we detected a MET amplification, the patient was treated with a combination of osimertinib and crizotinib without any obvious side effect. This combination led to a durable response with an excellent extra-cerebral control (Figure 4E) while also keeping the brain metastasis under control. This clinical case further validates the EGFR/MET mice as a preclinical model mimicking the human disease.

## 3. Discussion

Here, we described a new mouse strain in which human WT MET and the EGFR^T790M/L858R^ oncogenes are concomitantly expressed in the lung. These mice develop lung tumors and die due to breathing problems earlier than mice that only express EGFR^T790M/L858R^. This is despite the fact that in EGFR/MET mice, the number of full-blown adenocarcinomas was smaller, possibly because of a lack of time to acquire further mutations to promote disease progression. On the contrary, the tumor burden at the time of death by suffocation was very similar among both strains, hence we suggest that an accelerated tumor growth in EGFR/MET mice could be at the origin of the accelerated death compared to EGFR mice.

An interesting feature in EGFR/MET-driven tumors, compared with EGFR-driven tumors, was the concomitant strong reduction in pEGFR and total EGFR. Remarkably, their levels were increased after treatment with crizotinib alone, demonstrating that the inhibitory effect is promoted by MET activity. We are not aware of any previous published study on this effect. On the other hand, we found some studies showing the same finding in their figures, but without explicitly mentioning it because they were focused on other aspects, for instance, in one study using the human HCC877 cell line with spontaneous MET amplification [13]. Furthermore, the group of Pasi Jänne reported at AACR 2019 three patient-derived models with mutant EGFR and MET amplification and those models had less EGFR expression than their controls [14]. Here, we also unveiled MIG6 as one important player in the MET-induced decreased levels of EGFR. At this stage, we cannot rule out that, besides MIG6, other EGFR negative regulators could also play a role in the MET-induced inhibition of EGFR. In any case, we think that this is a very important finding because single treatment with anti-MET TKIs in patients with EGFR-mutated and MET-amplified lung cancer will restore EGFR activity. Moreover, we also found that osimertinib alone did not affect pERK activity and increased pAKT expression in EGFR/MET-driven tumors, which could provide pro-survival signals to the cancer cells even in the absence of pEGFR. Therefore, our preclinical data strongly suggest that the combination of osimertinib and a MET inhibitor should be always proposed to this subset of patients as for instance in the clinical case reported in Figure 4E. Our work is in accordance with previous clinical studies showing a poor response to crizotinib alone in patients with EGFR-mutated lung cancer with MET amplification [15].

Our preclinical model mimics the therapeutic responses to osimertinib and the combination of osimertinib and crizotinib, a MET TKI. Indeed, in EGFR/MET mice, this combination increased survival, decreased tumor burden and reduced the expression levels of all the tested key proteins implicated in lung adenocarcinoma (pEGFR, pMET, pAKT and pERK), compared with the single treatments or vehicle. Moreover, we observed a decrease in total MET and total EGFR levels. This is in accordance with previous work showing that when targeted therapy is effective in EGFR-mutated cancer cell lines, EGFR is downregulated [16], and further supports the effectiveness of the crizotinib and osimertinib combination in EGFR/MET-driven tumors. Finally, this combination was effective in one EGFR-mutated patient with MET amplification at our local hospital, Montpellier Cancer Institute, and fits with a very recent clinical trial showing that the combination of osimertinib and a MET inhibitor (savolitinib) has a therapeutic effect in patients with EGFR-mutated and MET-amplified lung cancer following relapse after osimertinib [5], further validating the EGFR/MET mouse as a preclinical model mimicking the human disease.

In summary, our new preclinical mouse model will help the lung cancer community to develop and test in vivo new therapeutic options for patients with EGFR-mutated lung cancer who relapse after osimertinib-based therapy due to the appearance of MET amplification, for instance, the bispecific antibody amivantamab against EGFR and MET [17].

## 4. Material and Methods

### 4.1. Mice and Genotyping

Tet-on-EGFR^T790M/L858R^, Tet-on-MET and CCSP-rtTA mice were described previously [18,19,20]. DNA was isolated using the REDExtract-N-Amp Tissue PCR Kit (Sigma) according to the manufacturer’s protocol. The rtTA, EGFR and MET transgenes were detected by PCR from DNA tails as described previously [18,19,20]. Transgenes were induced by using doxycycline-supplemented food (1 mg/kg) purchased from SAFE.

Animal procedures were performed according to protocols approved by the French National Committee of Animal Care.

### 4.2. Cell Culture, Western Blotting and Transfection Reagents

The H1975 and A549 lung cancer cell lines were from the American Type Tissue Culture Collection (ATCC) and were maintained in RPMI 1640 supplemented with 10% fetal bovine serum and antibiotics.

Western blotting was performed as previously described [21], with antibodies against pEGFR (#2234), EGFR (#4267), pMET (#3077), MET (#8198), pAKT (#4060), AKT (#4691), pERK (#4370), ERK (#9102) and MIG6 (#2440) from Cell Signaling Technology, tubulin (#4026) from Sigma, HSC70 (sc-7298) and MIG6 (sc-137154) from Santa Cruz. Secondary antibodies were horseradish peroxidase-linked anti-rabbit (#7074) and anti-mouse (#7076) IgG from Cell Signaling Technology. Tubulin served as a loading control for human samples and HSC70 for murine samples.

The siRNA non-targeting control (siNT, #D-001206-14) and that against MIG6 (siMIG6, #L-017016-001) from Dharmacon were used to transfect cells at 20 nM with Dharmafect1 reagent following the manufacturer’s instructions.

The full uncropped WB are available at Appendix A.

### 4.3. Generation of Cell Lines with Inducible MET Overexpression

MET cDNA was purchased from Addgene (#37560) and inserted into the pENTRY vector (Invitrogen) by PCR using the following forward and reverse primers: 5′-CACCATGAAGGCCCCCGCTGT-3′ and 5′-CTATGATGTCTCCCAGAAGGAGGC-3′. MET cDNA was then transferred into a pCLX vector using the Gateway system to generate the pCLX-METOX plasmid for lentiviral particle production in HEK 293T cells, following the manufacturer’s guidelines (Sigma, St. Louis, MO, USA). Target cells were infected with METOX-encoding lentiviruses supplemented with polybrene (8 mg/mL) and cultured in the presence of puromycin (2 mg/mL) for 4 days.

### 4.4. Treatments in Mice

Crizotinib (#AB-M1765) from AbMole and osimertinib (#T2490) from TargetMol were resuspended in vehicle (0.5% (w/v) methylcellulose, 0.2% Tween 80 (w/v)) and administered (50 mg/kg/day and 2.5 mg/kg/day, respectively) by oral gavage 5 days per week.

### 4.5. qPCR

For mRNA expression, total RNA was isolated using the RNeasy Mini Kit (Quiagen, Hilden, Germany). Complementary DNA was synthesized using SuperScriptIII (Invitrogen, Waltham, MA, USA) and qPCR was performed using the following primer pairs:

Human ERRFI1 forward: 5′-TGTGAACGGGGTTCTAGGC-3′ and reverse: 5′-TGGTCAGACACATAGCTGAGA-3′

Mouse Errfi1 forward: 5′-TGGCCTACAATCTGAACTCCC-3′ and reverse: 5′-GACCACACTCTGCAAAGAAGT-3′

Human ACTB forward: 5′-CTGTCTGGCGGCACCACCAT-3′ and reverse: 5′-GCAACTAAGTCATAGTCCGC-3′

Mouse Actb forward: 5′-ATGCTCTCCCTCACGCCATC-3′ and reverse: 5′-CACGCACGATTTCCCTCTCA-3′.

qPCR analyses were performed using SYBR Green (Applied Biosystems, Waltham, MA, USA) and the specific primers described above. Real-time PCR was carried out on a LightCycler 480 II (Roche, Basel, Switzerland). Reactions were run in triplicate. Expression data were normalized to the geometric mean of the housekeeping gene ACTB (Actb for mouse samples) to control the variability in expression levels and analyzed using the 2-DDCT method.

### 4.6. Analysis of Publicly Available Datasets

To analyze the relationship between pEGFR and pMET expression and between *EGFR*, *MET* and *ERRFI1* mRNA expression in lung adenocarcinomas, data were extracted from TCGA [9] through www.cbioportal.org (we downloaded the data on 26 April 2021). Specifically, on the home page of the website, the RPPA data as Z-score expression in the Lung Adenocarcinoma (TCGA, PanCancer Atlas) database from 360 patients were downloaded for pEGFR and pMET analysis, and the *EGFR* vs. *ERRFI1* and *MET* vs. *ERRFI1* mRNA expression Z-scores relative to all samples (log RNA Seq V2 RSEM) in the Lung Adenocarcinoma (TCGA, PanCancer Atlas) database from 510 patients were downloaded. The correlations between Z-scores were then analyzed by Pearson correlation and plotted using Prism GraphPad. Linear regression with 95% confidence intervals was also calculated and plotted using Prism GraphPad.

### 4.7. Histopathology and Immunohistochemistry

Lung lobes were fixed, embedded in paraffin and stained with hematoxylin and eosin (HE). To calculate tumor burden, both tumor area and total lung area were measured using ImageJ software and presented as percentages. For pathological analysis of HE-stained tissue sections, classical cytological and architectural features (e.g., invasion and high mitotic rate) were examined by our expert pathologist (M.C.), who also counted the number of adenomas and adenocarcinomas in each sample to show the percentage of adenocarcinomas.

### 4.8. Statistical Analysis

Unless otherwise specified, data are presented as mean ± standard error of the mean (SEM). One-way analysis of variance (ANOVA) was carried out to compare *ERRFI1* mRNA expression levels (Figure 3D and Figure 4D), and tumor burden among groups of mice receiving different treatments (Figure 3B and Figure 4B). An unpaired *t*-test was used to compare two different groups, as shown in Appendix A, Figure 2D,E and Appendix A. The Kaplan–Meier survival curves were compared with the Mantel–Cox test. All statistical analyses were performed using Prism GraphPad software.

### 4.9. Clinical Case

Camille Travert and Xavier Quantin, authors of the study, followed this patient at Montpellier Cancer Institute (ICM) in Montpellier. The patient’s data presented were collected after her oral consent. We used months of treatment instead of real dates to maintain confidentiality and the data presented are strictly limited to this study.

## 5. Conclusions

We have developed a new preclinical mouse model that will help the lung cancer community to develop and test in vivo novel therapeutic options for patients with EGFR-mutated lung cancer who relapse after osimertinib therapy due to the appearance of MET amplification.

## Figures and Tables

**Figure 1 cancers-13-03441-f001:**
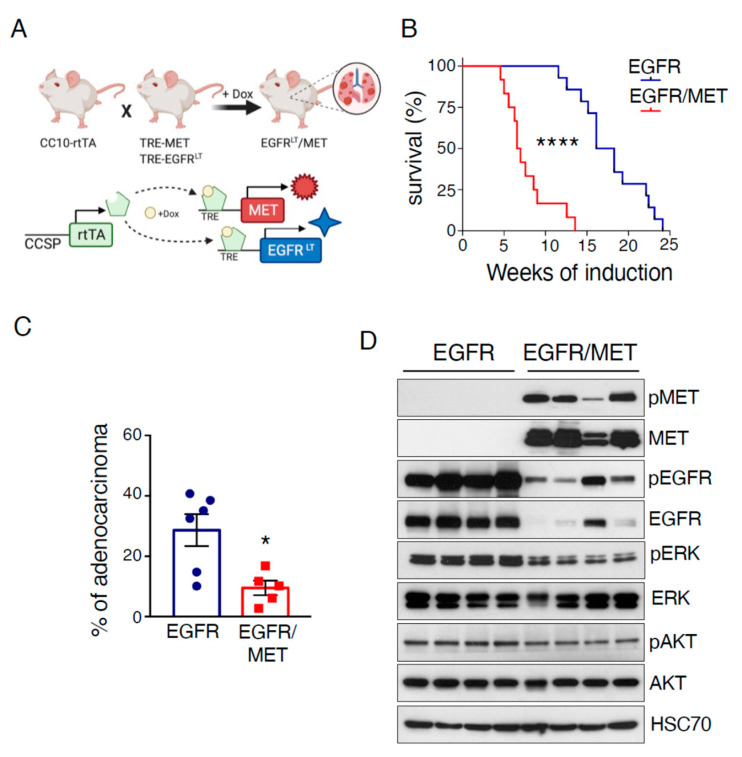
MET overexpression decreases survival of mice harboring EGFR-driven lung adenocarcinoma. (**A**) Cartoon depicting the strategy to generate EGFR/MET mice (top) and to activate the lung-specific expression of the two transgenes upon exposure to doxycycline diet (bottom). (**B**) Kaplan–Meier plot showing the survival rate of EGFR (*n* = 14) and EGFR/MET (*n* = 12) mice after doxycycline induction; **** *p* ≤ 0.0001 (Mantel–Cox test). (**C**) Percentage of adenocarcinomas vs. adenomas in each genotype. Values correspond to the mean ± SEM; * *p* ≤ 0.05 (unpaired *t*-test). EGFR (*n* = 6) and EGFR/MET (*n* = 5). (**D**) Immunoblotting of the indicated proteins in lung tumors from EGFR and EGFR/MET mice.

**Figure 2 cancers-13-03441-f002:**
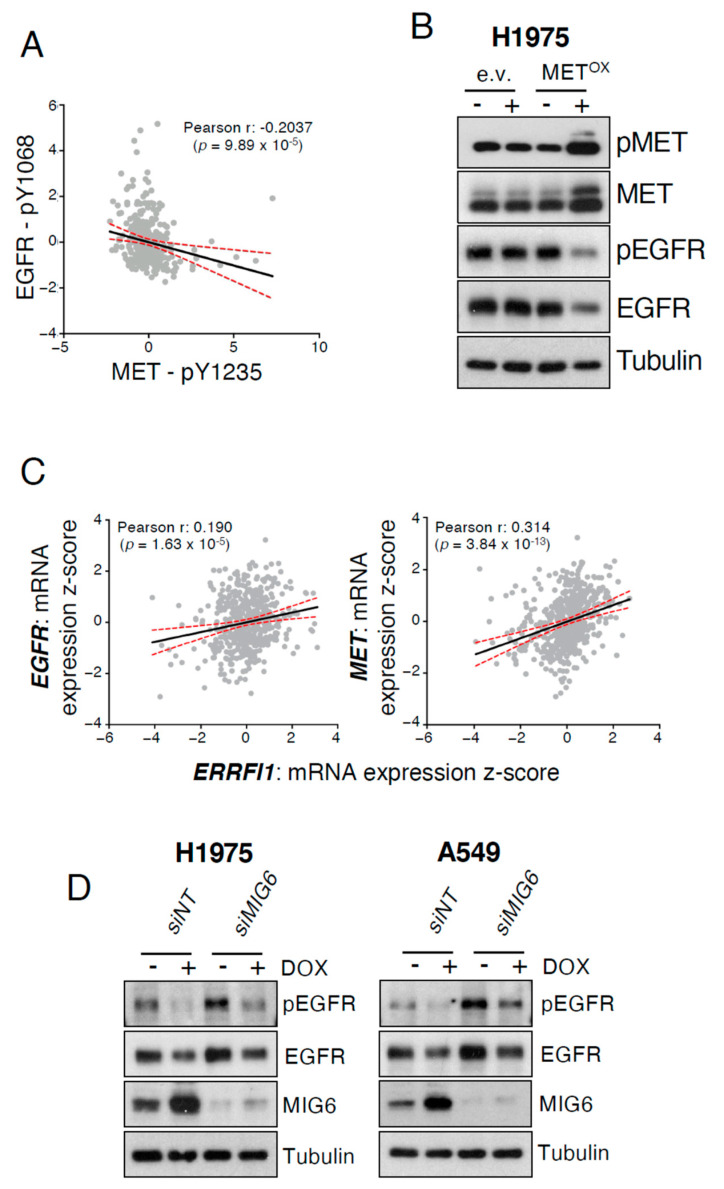
MET expression decreases EGFR activity. (**A**) Correlation analyses of pMET and pEGFR expression status (Z-scores) in 360 patients from the Lung Adenocarcinoma (TCGA, PanCancer Atlas) dataset. Data were analyzed by Pearson coefficient analysis (r = −0.2037). Linear regression (black line) and 95% confidence intervals (dashed red lines) are also indicated. (**B**) Immunoblotting of the indicated proteins in H1975 cells infected with lentiviral particles harboring the doxycycline-inducible MET construct (METOX) or empty vector (e.v.). Cells were incubated (+) or not (−) with 1 μg/mL of doxycycline for 48 h. This is a representative example of an experiment performed twice. (**C**) Correlation analyses for *EGFR* vs. *ERRFI1* (MIG6 gene) and *MET* vs. *ERRFI1* (MIG6 gene) mRNA expression (Z-scores) in 510 patients from the Lung Adenocarcinoma (TCGA, PanCancer Atlas) dataset. Z-scores were analyzed by Pearson coefficient analysis (r = 0.190 and r = 0.314, respectively). Linear regression (black line) and 95% confidence intervals (dashed red lines) are also indicated. (**D**) Immunoblotting of the indicated proteins in H1975 and A549 cells infected with lentiviral particles harboring the doxycycline-inducible MET construct (METOX) and transfected with non-targeting siRNA (*siNT*) or siRNA targeting MIG6 (*siMIG6*). Cells were incubated (+) or not (−) with 1 μg/mL of doxycycline for 48 h. This is a representative example of an experiment performed twice.

**Figure 3 cancers-13-03441-f003:**
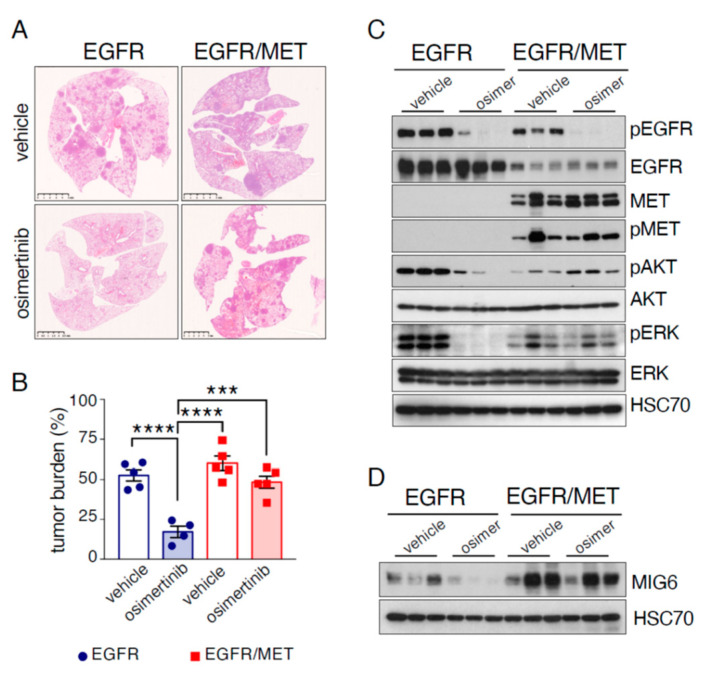
EGFR/MET-driven tumors are resistant to osimertinib. (**A**) Hematoxylin and eosin staining of whole lungs from EGFR mice treated with vehicle (*n* = 5) or osimertinib (*n* = 4) and EGFR/MET mice treated with vehicle (*n* = 5) or osimertinib (*n* = 5) for 4 weeks. Scale bar, 5 mm. (**B**) Tumor burden from same samples as described in (**A**). Values are the mean ± SEM; *** *p* ≤ 0.001, **** *p* ≤ 0.0001 (one-way ANOVA followed by Tukey post hoc test). (**C**) Immunoblotting of the indicated proteins in lung tumors from EGFR and EGFR/MET mice treated with vehicle or osimertinib. (**D**) Immunoblotting of the indicated proteins in lungs tumors from EGFR and EGFR/MET mice treated as in (**C**).

**Figure 4 cancers-13-03441-f004:**
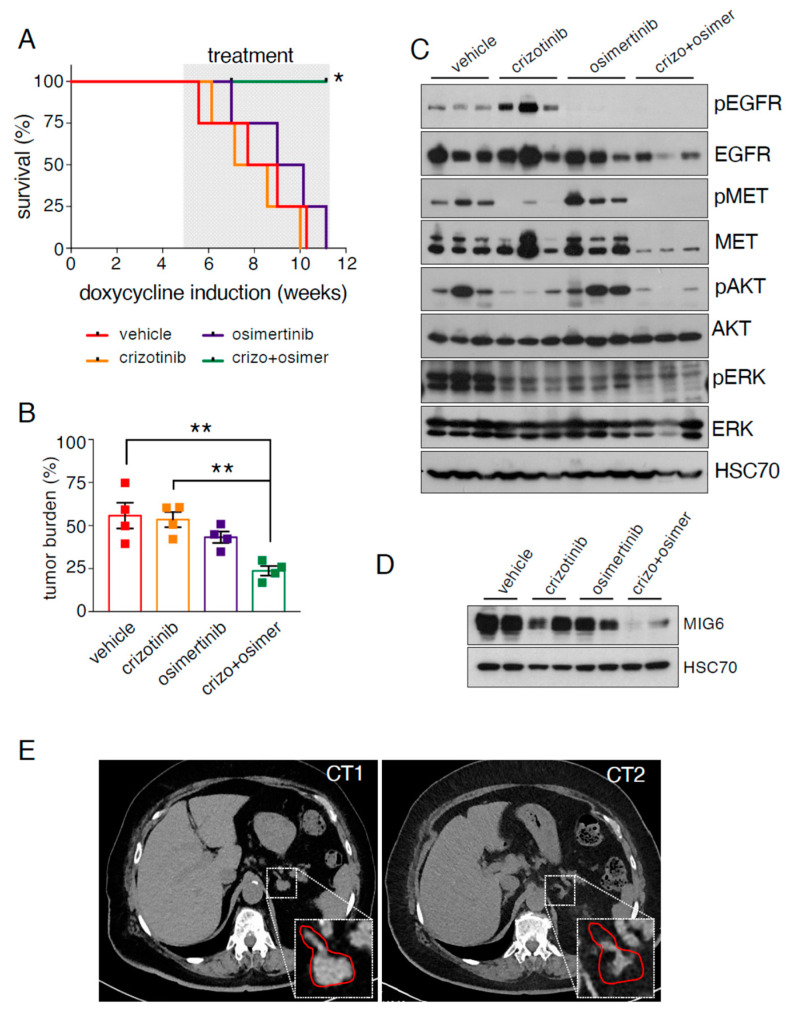
EGFR/MET-driven tumors are sensitive to the osimertinib and crizotinib combination. (**A**) Kaplan–Meier plot showing the survival rate of EGFR/MET mice treated with vehicle, crizotinib and/or osimertinib (*n* = 4 mice/group) for 6 weeks. The x axis shows the time since tumor induction by doxycycline; * *p* ≤ 0.05 (Mantel–Cox test) (**B**) Tumor burden from the mice described in (**A**) (*n* = 4 animals per treatment group). Data are the mean ± SEM; ** *p* ≤ 0.01 (one-way ANOVA followed by Tukey post hoc test). (**C**) Immunoblotting of the indicated proteins in lung protein extracts from the same mice described in (**A**). (**D**) Immunoblotting of the indicated proteins in lung protein extracts from the same mice described in (**A**). (**E**) CT scan of a patient presenting EGFR del19 mutation and MET amplification upon relapse after osimertinib single treatment (CT1) and 3 months after osimertinib and crizotinib treatment started (CT2). Magnification of the tumoral area is presented and highlighted in red.

## Data Availability

The clinical data presented in this study are available on request from the corresponding author. The data are not publicly available due to patient protection.

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
