# Peer review of "Generation and Characterization of a New Preclinical Mouse Model of EGFR-Driven Lung Cancer with MET-Induced Osimertinib Resistance"

_cancers, 2021, doi:10.3390/cancers13143441_

Round 1
Reviewer 1 Report
Major comments
In this study, authors generated a mouse model harboring the EGFR mutant and MET overexpression to mimic the MET amplification observed in lung cancer patients. Their findings showed that the survival of EGFR/MET mice was reduced as compared with EGFR mutant alone. In addition, the EGFR/MET-driven lung tumors were resistant to osimertinib, a third-generation anti-EGFR TKI, and osimertinib combined with crizotinib (an anti-MET TKI) improved survival and reduced tumor burden in the EGFR/MET mice. These characteristics are also observed in lung cancer patients. Generally, this study has merits and interests. There are some suggestions to improve this manuscript.
Crizotinib is also used in treatment of lung cancer patients harboring ALK mutant. What is the expression level of ALK in this model? In addition, does ALK play a role in the therapeutic effect of combination of osimertinib and crizotinib in this model?
line 60, 129-143, the simultaneous use of MIG6 and ERRFI1 is confusing.
line 153, “Cells were incubated (+) or not (-) with 1 g/ml of doxycycline for 48h.”, “1 g/ml” ?
line 248-249 “in EGFR/MET mice, the number of full-blown adenocarcinomas was smaller, …”, what makes the EGFR/MET mice die earlier than EGFR mice?
Reviewer 2 Report
Mancini et al generated a new preclinical mouse model od EGFR-driven lung cancer with MET overexpression and demonstrated the resistance to EGFR inhibition and the sensitivity to the combination of EGFR and MET inhibition. This model clearly will help further research focusing on the combination targeted therapy in the lung cancer field. They also found an interesting phenomenon that MET overexpression negatively regulated EGFR expression and activation. However, the role of MIG6 was not proven by their data. Additional experiments are necessary to claim the causality between MIG6 and EGFR or MET.
Major points
- The authors showed that the expression of ERFFI1 (MIG6) was upregulated by MET overexpression in the cell lines and the mouse models. To assess the biological function of MIG6, the overexpression and the knockdown or knockout of MIG6 is required.
- In Figure 2A, the plots of TCGA data showed two populations with either high phosphor-EGFR or high phospho-MET. The authors should describe these as mutually exclusive two populations rather than a general negative correlation.
Minor points
- Pasi Jänne’s group reported their patient-derived models of lung cancer with EGFR activating mutations and genomic MET copy number gain at AACR in 2019 (DOI: 10.1158/1538-7445.AM2019-1732). They described reduced EGFR expression in EGFR/MET-codependent models. Although these data have been not published, it supports the clinical relevance of the findings in this manuscript.
Round 2
Reviewer 2 Report
The authors responded to all comments by the reviewers appropriately and strengthened their findings by adding new data.